# GRAPH GENERATION VIA SCATTERING

## ABSTRACT

Generative networks have made it possible to generate meaningful signals such as images and texts from simple noise. Recently, generative methods based on GAN and VAE were developed for graphs and graph signals. However, the mathematical properties of these methods are unclear, and training good generative models is difficult. This work proposes a graph generation model that uses a recent adaptation of Mallat's scattering transform to graphs. The proposed model is naturally composed of an encoder and a decoder. The encoder is a Gaussianized graph scattering transform, which is robust to signal and graph manipulation. The decoder is a simple fully connected network that is adapted to specific tasks, such as link prediction, signal generation on graphs and full graph and signal generation. The training of our proposed system is efficient since it is only applied to the decoder and the hardware requirement is moderate. Numerical results demonstrate state-of-the-art performance of the proposed system for both link prediction and graph and signal generation. These results are in contrast to experience with Euclidean data, where it is difficult to form a generative scattering network that performs as well as state-of-the-art methods. We believe that this is because of the discrete and simpler nature of graph applications, unlike the more complex and high-frequency nature of Euclidean data, in particular, of some natural images.

## 1 INTRODUCTION

Generative neural networks have been successfully applied to various tasks such as the generation of images and texts. Their development is based on fruitful methods of deep learning, such as convolutional and recurrent neural networks. These and other methods of deep learning, which were initially developed for problems in the Euclidean domain[1], have been successfully generalized to address supervised learning tasks in the graph domain. In particular, a variety of graph convolutional networks have been developed, including networks with prescribed parameters (Zou & Lerman, 2018; Gama et al., 2018) and trained networks (Henaff et al., 2015; Defferrard et al., 2016; Kipf & Welling, 2017; Chen et al., 2017). It is natural to use these tools to build generative models in the graph domain.

Most generative graph networks directly use standard graph networks by following either of the following two generative frameworks: the generative adversarial network (GAN) (Goodfellow et al., 2014) and the variational auto-encoder (VAE) (Kingma & Welling, 2014). In a GAN, a generator and an auxiliary adversarial discriminator are trained together. On the other hand, in VAE, an encoder and a decoder (or generator) are both trained according to Bayesian models. Both frameworks contain two components (generator and discriminator or encoder and decoder), where each of them requires training. For GAN, training two components corresponds to a difficult min-max problem. On the other hand, training the two components in VAE can be described as a non-convex minimization problem. However, it is a crude approximation to the motivating variational inference formulation. Given an encoding process with guaranteed mathematical properties, one can focus on training only the decoder, or the generator, which is more tractable. In the Euclidean domain, Angles & Mallat (2018) use the scattering transform as an encoder, which is robust to deformations of input signals, and learn a generative model by minimizing the $l_1$-loss for reconstructing the training images. We adopt a similar method, using the graph scattering transform (Zou & Lerman, 2018) as an encoder for graph signals, which is robust to signal and graph manipulations, and train

---

[1] We remark that the Euclidean and graph domains include scenarios whose underlying datasets have Euclidean and graph structures, respectively.

neural networks corresponding to respective tasks. However, numerical experience indicates that a conventional generative scattering network is not as competitive as state-of-the-art results based on GAN and VAE. This is probably due to the complexity of some Euclidean-type datasets and their nontrivial high frequency components. Nevertheless, graph-type datasets have a discrete nature, and they often do not exhibit high frequency components. Therefore, the graph scattering transform might be competitive and efficient for specific generative tasks in the graph domain.

We consider three types of graph generation tasks:

- *Link prediction*: In this task, one is interested in predicting whether two vertices from the same graph are connected. The common input includes a graph with missing edges and features of vertices. The goal is to decide whether an edge exists between any pair of vertices. This can be viewed as generating a graph from a latent representation of the partially available graph.

  A well-known application of this task is the prediction of citations. Common citation datasets were collected by Sen et al. (2008) and further pre-processed by Kipf & Welling (2016). These datasets of publications and citations contain features for each publication as well as citation linkage, which are modeled by an undirected graph. In the pre-processed data, one only partially knows the citation linkage and the task is to recover the citations for all pairs of publications.

- *Signal generation on graphs*: In this task, the graph is fixed and the set of vertices and edges is known. The goal is to generate signals on the given graph. Although we are unaware of convincing data of this type, we believe that this is a potentially useful task. Among the three tasks we review here, it is most similar to generation tasks in Euclidean domains. We can thus enforce some graph structure in special Euclidean-type or grid-type datasets. Here we pursue this idea with the Fashion-MNIST dataset of images of clothing items (Xiao et al., 2017). We do not consider the domain of a $28 \times 28$ pixel image as Euclidean, but associate to it a graph, where each pixel is connected by an edge with its nearby neighbors. Each image is then a signal on this graph. One then needs graph-based methods for generating these signals.

- *Graph and signal generation*: In this task, one needs to generate both the graph structure and the signals on the graph. An interesting application is the design of chemical molecules. A network learns from a given dataset, such as QM9 (Ramakrishnan et al., 2014), to generate both the atoms (signals on vertices) and the bonds (edges) from a latent sample. This can be used as a pure machine learning approach for the design of new drugs (Olivecrona et al., 2017).

Our proposed method is easy to implement. Furthermore, the adjustment of the structure of the decoder to the three types of tasks does not require a lot of effort. Unlike GAN or VAE, the model in this paper does not require training two components either iteratively or at the same time. Meanwhile, there is flexibility in designing the graph wavelets and in choosing the decoder structure. We believe it is a highly adaptable method for various graph tasks and indeed our numerical experiments demonstrate competitive results.

## 2 BACKGROUND

We overview previous relevant works as follows: §2.1 reviews generative scattering networks, §2.2 reviews graph convolutional networks, and §2.3 reviews some recent graph generative models.

### 2.1 SCATTERING NETWORKS FOR GENERATIVE MODELS

The generative scattering network (Angles & Mallat, 2018) can be considered as an encoder-decoder system in which one only needs to train the decoder. The feature extraction part of the encoder is a scattering transform (Mallat, 2012; Bruna & Mallat, 2013) with fixed parameters. It provides multi-scale signal representation, which is Lipschitz continuous with respect to small deformations. The next part of the encoder transforms the representations of the input signals into samples of a Gaussian latent variable. Figure 1 shows the structure of a generative scattering model.

The decoder of the generative scattering network can be taken to be a multi-layer perceptron (MLP). We denote the encoder by $\Phi$ and the decoder by $D$. Given input samples $\{\boldsymbol{x}^{(t)}\}_{t=1}^{T}$, the decoder is

trained by minimizing the expected reconstruction loss

$$L_\Phi(D) = \sum_{t=1}^{T} \left\| \boldsymbol{x}^{(t)} - D(\Phi(\boldsymbol{x}^{(t)})) \right\|_1 .$$

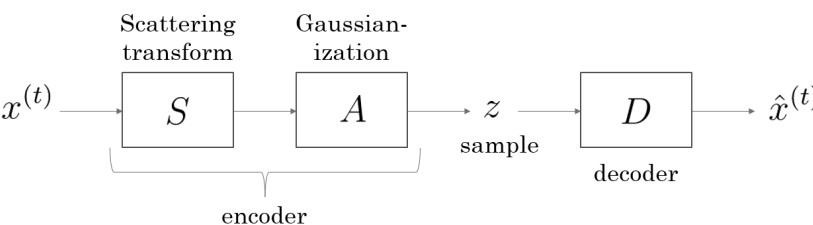

Figure 1: The structure of a generative scattering network.

This generative scattering network is interesting because of its simple implementation with a prescribed encoder and because of the mathematically guaranteed properties of this encoder.

## 2.2 Graph convolutional networks

Convolution is a key contributor for the recent success of deep learning. In the Euclidean domain, convolutional networks are helpful in learning multi-scale representations. The same idea was introduced to graphs by exploiting the spectral graph representation, that is, the spectral decomposition of the graph weight matrix or graph Laplacian (Bruna et al., 2014; Henaff et al., 2015). A common proposal for a graph convolution uses pointwise multiplication of the graph Fourier-transformed signals, where the graph Fourier transform uses the basis of the spectral decomposition of the Graph Laplacian in place of the discrete Fourier basis. One can apply nonlinear functions such as the ReLU for each graph vertex. A variety of good approximations to the spectral approach are able to speed up the spectral decomposition process and maintain accuracy (Defferrard et al., 2016; Kipf & Welling, 2017).

A special type of graph convolutional network (GCN) is the graph scattering network (Zou & Lerman, 2018), which does not require training and was proved to be approximately invariant to permutations and stable to sufficiently small signal or graph manipulations. A graph scattering network uses graph wavelets (Hammond et al., 2011) defined on the eigenspace of the graph Laplacian to construct multi-layer models. Alternatively, Gama et al. (2018) construct graph scattering transforms using an earlier graph wavelet transform (Coifman & Maggioni, 2006). In general, the graph wavelets of Hammond et al. (2011) are more flexible as one can choose different kinds of wavelets on the spectral domain according to different tasks. While the graph wavelets of Coifman & Maggioni (2006) are not flexible, they use the normalized graph Laplacian and the corresponding diffusion map and metric, which might be natural for particular applications.

## 2.3 Graph generative networks

Several recent papers address graph generation networks with either a GAN or a VAE structure. We briefly survey these two types of graph models.

### 2.3.1 GAN-type graph networks

As in regular GAN's, GAN-type graph networks use a discriminator in order to compete with the generator and make its training more powerful. In the following, we describe some recent graph generation models of this type.

**NetGAN.** The NetGAN (Bojchevski et al., 2018) first generates random walks on the graph vertices, whose features are given. Such features include various properties of the vertices. Specifically, the random walks are sequential processes that depart from the graph vertices and travel along edges of the complete graph. To form the edges and thus generate the graph, a threshold is put on the

frequency of travels between pairs of vertices. A Wasserstein GAN structure is used for training the generator. Both the generator and the discriminator take the structure of a Long-Short-Term-Memory (LSTM) network (Hochreiter & Schmidhuber, 1997). This model is mainly designed for link prediction or graph generation with given features of vertices. It is successful in generating graphs with properties, such as max degree and triangle count, that are similar to those of the samples. However, the model is complicated and hard to train.

**GraphGAN.** The GraphGAN (Wang et al., 2018) generates for each vertex the distribution of edges associated with it. It adopts the original GAN setting, where the objective is to minimize the sum over all vertices of KL-divergences between real and generated distributions. An interesting graph-type softmax function is used to overcome some problems that arise when applying the common softmax function to this network. This model is used for link prediction problems or graph generation with given features of vectors. Wang et al. (2018) demonstrate results for node classification and recommendation problems, whose tasks are very similar to that of the link prediction problem.

**MolGAN.** The MolGAN (De Cao & Kipf, 2018) generates signals with respect to both vertices and edges. It seeks to minimize the sum of two Wasserstein loss functions, where the first one includes a sum over all vertices and the second one a sum over pairs of vertices to account for possible edges. Therefore, MolGAN is designed to solve the graph and signal generation problem in the context of molecule generation. The signals for edges provide the bonds and the vertex signals assign for each vertex a type of atom. In order to emphasize respective molecular properties and improve scores for them, MolGAN utilizes an auxiliary reward network. The generator is trained with both the adversarial discriminator and the reward network.

### 2.3.2 VAE-TYPE GRAPH NETWORKS

VAE can be generalized for graphs by considering distributions on graph vertices. It still utilizes the encoder-decoder system, but the encoder is usually taken to be a graph convolutional neural network.

**VGAE.** The variational graph auto-encoder (VGAE) of Kipf & Welling (2016) is designed mainly for link prediction. Its generator has a structure similar to a regular VAE. It contains a GCN (Kipf & Welling, 2017) encoder and an inner product decoder. The GCN produces either the mean and variance of the latent code (in the probabilistic version), or just the latent code itself. The latent code produces vectors for the vertices, and their inner-product provides a similarity measure, which is used to determine whether a link exists between two vertices.

**GraphVAE.** The GraphVAE (Simonovsky & Komodakis, 2018) also uses a VAE setting. Unlike VGAE, it focuses on the graph and signal generation problem. To achieve this, it considers the adjacency matrix $A$, edge features $E$, which is a set of signals of the graph edges, and vertex features $F$, which is a set of signals on the graph vertices. The loss that needs to be optimized is a function of $A$, $E$ and $F$. The computational complexity of GraphVAE is of order $O(N^4)$. Therefore, it is most suitable for tasks with a small number of vertices, such as molecule generation. A recent variant of VAE, JT-VAE (Jin et al., 2018), is specifically designed for generating complex molecules while enforcing validity. Since our numerical experiments do not consider complex molecules, it is not relevant for this manuscript.

## 3 GRAPH GENERATIVE SCATTERING NETWORK

In general, the design of the graph generation networks in §2.3 is complex and their hyperparameter selections are difficult. Furthermore, training GAN and VAE are known to be difficult. In view of these obstacles, we propose here the graph generative scattering network. It is composed of two components: an encoder and a decoder. The encoder is a graph scattering network (Zou & Lerman, 2018), which produces a latent representation for the graph signal. We use it to form a latent Gaussian distribution. The parameters of the graph scattering network are predetermined, unlike the parameters of a generative auto-encoder that are learned. The decoder learns the mapping from the latent Gaussian distribution according to the corresponding tasks. In any of the graph-specific tasks we mentioned in §1, the decoder can be taken to be a network with fully-connected layers, whose structure is determined by the specific task. More details on forming the encoder and decoder are provided in §3.1 and §3.2, respectively.

### 3.1 Details of the Encoder

In order to fully understand the formation of the encoder, we review the graph scattering network of Zou & Lerman (2018) and explain how to form a Gaussian distribution from its output. We consider a graph $G = (V, E)$ with $|V| = N$ vertices. A signal in $L^2(V; \mathbb{R}^K)$ can be regarded as a matrix $\boldsymbol{X} \in \mathbb{R}^{N \times K}$. The scattering transform can be regarded as a function that acts on the columns of $\boldsymbol{X}$. Let $\boldsymbol{L} \in \mathbb{R}^{N \times N}$ be the unnormalized graph Laplacian $\boldsymbol{L} = \boldsymbol{D} - \boldsymbol{W}$, where $\boldsymbol{D}$ is the diagonal matrix of degrees and $\boldsymbol{W}$ is the weight (adjacency) matrix whose $(n, m)$-th entry is the weight of the edge connecting vertices $v_n$ and $v_m$. Its spectral decomposition can be written as $\boldsymbol{L} = \sum_{l=0}^{N-1} \lambda_l \boldsymbol{u}_l \boldsymbol{u}_l^*$, with $0 = \lambda_0 \leq \cdots \leq \lambda_{N-1}$. We assume dyadic wavelets $\phi$ and $\psi$ satisfying for some $J \in \mathbb{Z}$, $|\hat{\phi}_{-J}|^2 + \sum_{j > -J} |\hat{\psi}_j|^2 = 1$, where

$$\hat{\psi}_j(\omega) = \hat{\psi}(2^{-j}\omega) \text{ for } j > -J \text{ and } \hat{\phi}_{-J}(\omega) = \hat{\phi}(2^{-J}\omega).$$

For $\boldsymbol{f} \in \mathbb{R}^N$, the graph wavelet transform (Hammond et al., 2011) is

$$\boldsymbol{f} * \boldsymbol{\psi}_j = \sum_{l=0}^{N-1} \boldsymbol{u}_l \boldsymbol{u}_l^* \boldsymbol{f} \hat{\psi}(2^{-j}\lambda_l), \text{ for } j > -J; \text{ and } \boldsymbol{f} * \boldsymbol{\phi}_{-J} = \sum_{l=0}^{N-1} \boldsymbol{u}_l \boldsymbol{u}_l^* \boldsymbol{f} \hat{\phi}(2^J \lambda_l).$$

For any $m$ no larger than the number of layers, a path $p = (j_1, \cdots, j_m)$ is a vector of $m$ scales of the graph wavelets, which satisfy $0 \leq j_1, \cdots, j_m \leq J - 1$. The scattering propagator with respect to a path $p$ is obtained by consecutive application of convolutions with wavelets of these scales and absolute values, which serve as nonlinearities, in the following way

$$\boldsymbol{U}[p]\boldsymbol{f} = \left|\left|\left|\boldsymbol{f} * \boldsymbol{\psi}_{j_1}\right| * \boldsymbol{\psi}_{j_2}\right| * \cdots * \boldsymbol{\psi}_{j_m}\right|.$$

The scattering transform with respect to the path $p$ is obtained by $\boldsymbol{S}[p]\boldsymbol{f} = \boldsymbol{U}[p]\boldsymbol{f} * \boldsymbol{\phi}_{-J}$.

Let $\mathcal{P}$ denote the collection of all paths of length no larger than the number of layers. The scattering transform of $\boldsymbol{f}$ with respect to $\mathcal{P}$ is

$$\boldsymbol{S}[\mathcal{P}]\boldsymbol{f} = (\boldsymbol{S}[p]\boldsymbol{f})_{p \in \mathcal{P}}.$$

We note that the scattering transform depends on the underlying graph. For simplicity, we exclude this dependence from our notation.

For the $K$-dimensional signal $\boldsymbol{X} = [\boldsymbol{X}_1 | \cdots | \boldsymbol{X}_K] \in \mathbb{R}^{N \times K}$, the scattering transform is

$$\boldsymbol{S}[\mathcal{P}]\boldsymbol{X} = (\boldsymbol{S}[\mathcal{P}]\boldsymbol{X}_k)_{k=1}^K.$$

We remark that if the set $\mathcal{P}$ ~~is finite and~~ has $L$ elements, then denoting $M = LK$, $\boldsymbol{S}[\mathcal{P}](\boldsymbol{X})$ can be represented as a matrix in $\mathbb{R}^{N \times M}$ or a vector in $\mathbb{R}^{N \cdot M}$. We denote the latter vector by $\bar{\boldsymbol{X}}$.

To make an effective encoder, an affine transformation $\boldsymbol{A}$ is applied to $\boldsymbol{S}[\mathcal{P}](\boldsymbol{X})$ as a whitening filter as follows. Let $\{\bar{\boldsymbol{X}}^{(t)}\}_{t=1}^T$ be the scattering transform vectors of the input samples $\{\boldsymbol{X}^{(t)}\}_{t=1}^T$. Let $\mathcal{X}$ and $\bar{\mathcal{X}}$ be the representing matrices of $\{\boldsymbol{X}^{(t)}\}_{t=1}^T$ and $\{\bar{\boldsymbol{X}}^{(t)}\}_{t=1}^T$, respectively. That is, $\mathcal{X} = (\boldsymbol{X}^{(t)})_{t=1}^T \in \mathbb{R}^{T \times NK}$ and $\bar{\mathcal{X}} = (\bar{\boldsymbol{X}}^{(t)})_{t=1}^T \in \mathbb{R}^{T \times NM}$. As advocated in Angles & Mallat (2018), a dimension reduction by PCA can be further applied to $\{\bar{\boldsymbol{X}}^{(t)}\}_{t=1}^T$ and $\bar{\mathcal{X}}$. Next, using the following mean and sample covariance of the scattering transform vectors

$$\boldsymbol{\mu} = \frac{1}{T} \sum_{t=1}^T \bar{\boldsymbol{X}}^{(t)} \text{ and } \boldsymbol{\Sigma} = \frac{1}{T} \sum_{t=1}^T (\bar{\boldsymbol{X}}^{(t)} - \boldsymbol{\mu})(\bar{\boldsymbol{X}}^{(t)} - \boldsymbol{\mu})^*,$$

$\bar{\mathcal{X}}$ is linearly transformed in the following way

$$\boldsymbol{A}\bar{\mathcal{X}} = \boldsymbol{\Sigma}^{-1/2}(\bar{\mathcal{X}} - \boldsymbol{\mu}). \tag{1}$$

We denote the matrix $\boldsymbol{A}\bar{\mathcal{X}}$ by $\boldsymbol{\Phi}[\mathcal{P}](\mathcal{X})$, or in short $\boldsymbol{\Phi}(\mathcal{X})$. We note that the mapping $\boldsymbol{\Phi}$ is a decomposition of the scattering transform and whitening and can also be applied to any signal $\boldsymbol{X} \in \mathbb{R}^{N \times K}$. We denote the feature vector corresponding to the $K$-dimensional signal $\boldsymbol{X}$ by $\boldsymbol{z} = \boldsymbol{\Phi}[\mathcal{P}](\boldsymbol{X}) \in \mathbb{R}^{N \times M}$. We also refer to $\boldsymbol{z}$ as a latent code.

Zou & Lerman (2018) proved that the scattering transform $\boldsymbol{S}[\mathcal{P}]$ is robust to small perturbations of the signal and the graph. The affine transform $\boldsymbol{A}$ only amplifies their perturbation estimates by at most $\|\boldsymbol{\Sigma}^{-1/2}\|$, that is, by the spectral norm of $\boldsymbol{\Sigma}^{-1/2}$. Therefore, their result applies to the encoder.

## 3.2 DETAILS OF THE DECODER

We recall that the decoder is a network with fully-connected layers. We describe its architecture according to the following three different tasks.

### 3.2.1 LINK PREDICTION

For link prediction, we encode the features of the partially available graph into a latent vector, and use the same vector to generate the full graph via the learned decoder. Note that in this task only one fixed graph is given, and thus no Gaussianization procedure is applied in the encoder. That is, the linear transformation $\boldsymbol{A}$ in (1) is the identity. The input includes a weight matrix $\boldsymbol{W}_{\text{train}}$, which contains weights for the partially available edges, and a feature matrix $\boldsymbol{X} \in \mathbb{R}^{N \times K}$ of $K$-dimensional signals on the $N$ nodes. The encoder is a scattering network $\boldsymbol{\Phi}$ that maps $\boldsymbol{X}$ and $\boldsymbol{W}_{\text{train}} \in \mathbb{R}^{N \times N}$ into a latent representation $\boldsymbol{z} \in \mathbb{R}^{N \times M}$. As in Kipf & Welling (2016), the decoder is a simple network $\boldsymbol{D}$ such that $\boldsymbol{D}(\boldsymbol{z}) = \sigma(\tilde{\boldsymbol{D}}(\boldsymbol{z})\tilde{\boldsymbol{D}}(\boldsymbol{z})^T)$, where $\sigma$ is the softmax function and $\tilde{\boldsymbol{D}}$ is an MLP. The network $\boldsymbol{D}$, whose parameters are those of $\tilde{\boldsymbol{D}}$, is trained to minimize the cross-entropy loss function

$$L(\boldsymbol{D}) = \sum_{i,j:\boldsymbol{W}(i,j)\neq 0} \left[ -\log \boldsymbol{D}(\boldsymbol{\Phi}(\boldsymbol{X}, \boldsymbol{W}))(i,j) \right]. \tag{2}$$

The structure of the entire network is illustrated in Figure 2.

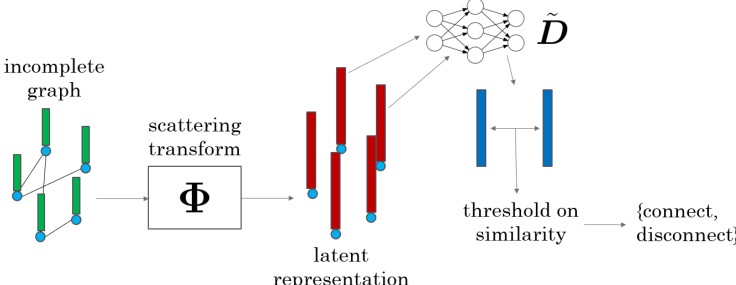

Figure 2: Sketch of a graph scattering network for link prediction.

### 3.2.2 SIGNAL GENERATION ON GRAPHS

For signal generation on graphs, which we also refer to as graph signal generation, one is given a fixed graph domain and different signals on the nodes of this graph and the goal is to generate similar signals. An input random variable $\boldsymbol{X} \in \mathbb{R}^{N \times K}$ is first mapped by the scattering transform to $\boldsymbol{S}[\mathcal{P}](\boldsymbol{X})$ and then to a Gaussian $\boldsymbol{z} = \boldsymbol{\Phi}[\mathcal{P}](\boldsymbol{X}) \in \mathbb{R}^{N \times M}$. The decoder $\boldsymbol{D}$ is taken to be an MLP that maps $\boldsymbol{z}$ to a matrix $\boldsymbol{D}(\boldsymbol{z})$ in $\mathbb{R}^{N \times K}$. The parameters of $\boldsymbol{D}$ are obtained by minimizing the reconstruction loss function

$$L(\boldsymbol{D}) = T^{-1} \sum_{t=1}^{T} \left\| \boldsymbol{X}^{(t)} - \boldsymbol{D}(\boldsymbol{\Phi}(\boldsymbol{X}^{(t)})) \right\|, \tag{3}$$

where $\{\boldsymbol{X}^{(t)}\}_{t=1}^{T}$ are the training data points. The structure of the generative network is the same as in Figure 1, where in the current case $S$ is the graph scattering transform. Figure 3 illustrates the generation procedure.

### 3.2.3 GRAPH AND SIGNAL GENERATION

The scattering transform can be used as an encoder for generating both the graph and the signal on it. We train two MLP's $\boldsymbol{D}_1$ and $\boldsymbol{D}_2$, where both take the Gaussian random variable $\boldsymbol{z} = \boldsymbol{\Phi}(\boldsymbol{X})$ as input. The network $\boldsymbol{D}_1$ is used to generate the graph signal $\boldsymbol{X}$ and the network $\boldsymbol{D}_2(\boldsymbol{z}) = \sigma(\tilde{\boldsymbol{D}}_2(\boldsymbol{z})\tilde{\boldsymbol{D}}_2(\boldsymbol{z})^T)$ is used to generate the weight matrix $\boldsymbol{W}$. They are trained at the same time, with the reconstruction loss function

$$L(\boldsymbol{D}_1, \boldsymbol{D}_2) = T^{-1} \sum_{t=1}^{T} \left[ \left\| \boldsymbol{W}^{(t)} - \boldsymbol{D}_1(\boldsymbol{\Phi}(\boldsymbol{X}^{(t)})) \right\| + \left\| \boldsymbol{X}^{(t)} - \boldsymbol{D}_2(\boldsymbol{\Phi}(\boldsymbol{X}^{(t)})) \right\| \right], \tag{4}$$

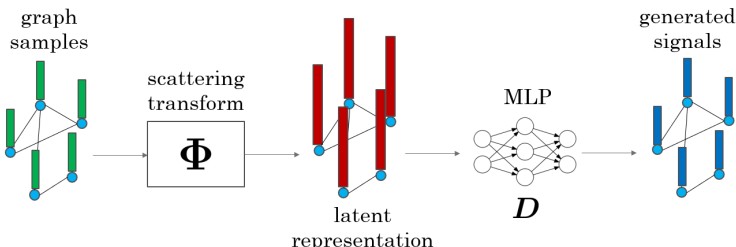

Figure 3: Sketch of a graph scattering network for signal generation on graphs.

where $\{(\boldsymbol{X}^{(t)}, \boldsymbol{W}^{(t)})\}_{t=1}^{T}$ are the training data points. The norms can be replaced with cross-entropy losses if one wants the outputs to be categorical. Figure 4 illustrates this generation procedure.

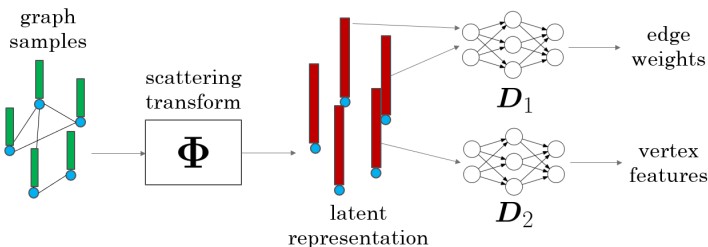

Figure 4: Sketch of the graph scattering network for graph and signal generation.

## 4 EXPERIMENTS

We test our proposed method, which we refer to in short as "SCAT", and compare it with other available algorithms using datasets addressing the three different tasks reviewed in §1. We describe our results in the three subsections below according to these tasks. Our method requires moderate memory and GPU, and we thus use a PC with 8GB RAM and a single GTX1060 GPU. All the neural networks in this section are trained using the TensorFlow library. Following Zou & Lerman (2018) we use in all experiments the simple Shannon wavelet, $J = 3$ and 3 layers.

### 4.1 LINK PREDICTION FOR CITATION DATA

We predict links for the three citation datasets of Sen et al. (2008): Cora, Citeseer and Pubmed. Each dataset contains information about publications in certain fields and the corresponding citation linkage between these publications. This information can be embedded in a graph in which the publications and citations are represented by vertices and edges, respectively. Although the citation link is directed, we follow the common convention of assuming an undirected and unweighted graph. That is, if any of two papers cite the other, an edge is drawn between them. Some characteristics of the three datasets are listed in Table 1.

Table 1: Characteristics of three citation datasets

| Dataset | Vertices | Edges | Classes | Features |
|---------|---------|-------|---------|----------|
| Citeseer | 3327 | 4732 | 6 | 3703 |
| Cora | 2708 | 5429 | 7 | 1433 |
| Pubmed | 19717 | 44338 | 3 | 500 |

We use the preprocessed data of Kipf & Welling (2016) (see https://github.com/tkipf/gae). It selects only a fraction of the original edges and proposes specific choices for training, validating and testing edges. The validation and test sets contain 5% and 10% of the total number of links, respectively. The use of only a fraction of the edges results in only partial knowledge of the citation linkage.

We construct the encoder of SCAT using a scattering transform, where the output dimension is reduced to 128. The decoder $\tilde{D}$ in §4.1 is taken to be a single dense layer of size 512, activated by the ReLU function. In order to minimize (2), we use the Adam optimizer with a learning rate of 0.001, where we train 1,000 epochs for each run. We record the following two common scores: area under curve (AUC) and average precision (AP). Similarly to Kipf & Welling (2016), we take 10 runs for each setting and record the average and standard deviation. Table 2 reports our results for SCAT together with results obtained in Kipf & Welling (2016) for VGAE, spectral clustering (SC) (Tang & Liu, 2011) and deep walk (DW) (Peyré et al., 2017) . We could test the results of VGAE using Cora and Citeseer on our own machine and thus reported these results, where we mention the corresponding results of Kipf & Welling (2016) in the caption of Table 2. However, on the high-dimensional Pubmed dataset, a direct run of the VGAE codes exhausts the computing resource on our machine. We also did not verify the reported results of SC and DW as we lacked their codes. We note that SCAT improves over the previous results for the three datasets. On average, for Cora/Citeseer/Pubmed, the scattering transform takes 11.85s/4.36s/156.23s while training each epoch takes 13.9ms/13.4ms/55.1ms. In comparison, we note that the training in VGAE for Cora/Citeseer takes 80.0ms/108.2ms for each epoch, but there is no initial scattering step.

Table 2: Results for link prediction using common citation datasets. Only the results of SCAT and VGAE for Cora and Citeseer were done on our machine. The rest are from Kipf & Welling (2016). We remark that for VGAE, Kipf & Welling (2016) reported the following AUC and AP for Cora: $91.4 \pm 0.01$ and $92.6 \pm 0.00$ and the following ones for Citeseer: $90.8 \pm 0.02$ and $92.0 \pm 0.00$.

| Dataset | Cora | | Citeseer | | Pubmed | |
|---|---|---|---|---|---|---|
| | AUC (%) | AP (%) | AUC (%) | AP (%) | AUC (%) | AP (%) |
| **SCAT** | $97.0 \pm 0.12$ | $96.6 \pm 0.11$ | $97.5 \pm 0.09$ | $97.3 \pm 0.06$ | $97.3 \pm 0.06$ | $97.0 \pm 0.03$ |
| VGAE | $91.7 \pm 0.28$ | $93.1 \pm 0.26$ | $91.2 \pm 0.08$ | $91.5 \pm 0.10$ | $96.4 \pm 0.00$ | $96.5 \pm 0.00$ |
| SC | $84.6 \pm 0.01$ | $88.5 \pm 0.00$ | $80.5 \pm 0.01$ | $85.0 \pm 0.01$ | $84.2 \pm 0.02$ | $87.8 \pm 0.01$ |
| DW | $83.1 \pm 0.01$ | $85.0 \pm 0.00$ | $80.5 \pm 0.02$ | $83.6 \pm 0.01$ | $84.4 \pm 0.00$ | $84.1 \pm 0.00$ |

## 4.2 SIGNAL GENERATION ON GRAPHS

We use the Fashion-MNIST dataset (Xiao et al., 2017) for a sanity check of SCAT for the problem of graph signal generation. Any element of this dataset is a $28 \times 28$ grayscale pixel image and can be considered as a graph in the following way: the pixels are the graph vertices, and nearby pixels are connected by graph edges. The edges and weights are formed as in Zou & Lerman (2018). That is, each pixel represents a vertex and it is connected with its four nearest neighbors with weight $e^{-1}$ and its four nearest diagonal neighbors with weight $e^{-2}$.

The encoder of this network is a graph scattering transform, whereas the output dimension is reduced to 256 from $28 \times 28 \times 13 = 10,192$. The decoder is an MLP of two hidden layers of size 512. In order to minimize (3), we use the Adam optimizer with a learning rate of 0.001, where we train 200 epochs for each run. We have restricted the dataset to the "boots" category, which contains 5,454 training examples. Sample images from this category are demonstrated in Figure 5a.

To generate an "image" (or graph signal), we take a sample $z \in \mathbb{R}^N$ from the standard normal distribution, and report the output of the decoder. Figure 5c illustrates samples generated by the network, while using the graph associated with a $28 \times 28$ pixel image. The reconstruction loss defined in (3) is 0.022. The relative error, which is obtained by dividing the reconstruction loss by the average L1-norm of the training images, which equals $174.56$, is $0.0126\%$. We remark that the scattering transform for the whole training data takes 5.05s and it takes 5.38s to train each epoch. In Figure 5b the encoder uses the original data of Figure 5a as input and its output is fed into the decoder. This figure then reports the corresponding output of the decoder.

We note that the random generated "images" (or graph signals) in Figure 5 cover a variation of boot shapes and heel types. However, some high frequency components of these images are missing. This is expected in scattering networks and is evident in results obtained by the Euclidean generative scattering network of Angles & Mallat (2018). Nevertheless, graph-type data often do not have high resolution and frequencies. For instance, in the molecular data reviewed in §4.3, the signals on each vertex just take five different values. For comparison, Appendix §6.1 shows generated graph-based images by a graph version of vanilla VAE and GAN. These results also indicate some problems with

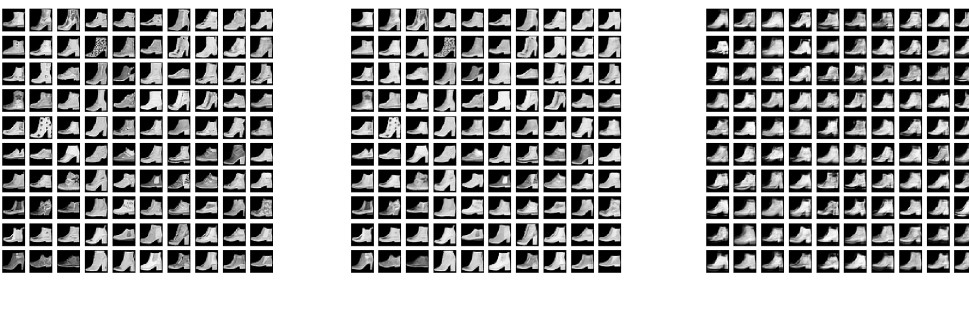

| (a) Original data. | (b) Data-based generation. | (c) Random generation. |

Figure 5: Original and generated images for boots of MNIST-fashion data. We assume the graph associated with the $28 \times 28$ pixel image. Figure 5a shows samples from the original data, 5b shows samples generated using features encoded by the scattering transform from the same samples, and Figure 5c shows samples generated from random Gaussian noise.

generating high-frequency components. This problem is more noticeable with our graph version of GAN than with the other two methods, that is, graph version of vanilla VAE and SCAT.

### 4.3    GRAPH GENERATION FOR MOLECULAR DATA

We test graph and signal generation using the molecular dataset QM9 (Ramakrishnan et al., 2014). This dataset contains 134k molecules made of the following atoms: C, H, O, N, and F. There are two common ways of embedding these kinds of datasets into an interpretable feature space. Kusner et al. (2017) treats molecules as "words" by looking at their simplified molecular-input line-entry (SMILE) strings. Graphs are also commonly used to represent molecules, where the graph vertices represent the atoms composing the molecule and the graph edges are the bonds. The vertex signals assign the four different atom types to the vertices. While there are five type of atoms, H is automatically determined by the other atoms and the given chemical bonds. We thus represent the atoms C, O, N and F as one-hot vectors, that is, by the four unit vectors in $\mathbb{R}^4$.

For graph generation, there is no unique benchmark for checking the quality of the generated graphs. Bojchevski et al. (2018) proposed to use graph properties such as the max degree and the number of triangles for graph generation. However, it is often hard to compare these graph properties and for molecular generation these ideas are not well-motivated. Samanta et al. (2018) proposed to check validity (whether a sample is a valid chemical molecule), uniqueness (whether a sample is unique among all generated samples) and novelty (whether a sample is different from any sample in the training data). We use this measure since it is more quantitative and experiments on QM9 by Simonovsky & Komodakis (2018) and De Cao & Kipf (2018) also report these quantities.

The training set of our main experiment uses 90k molecules composed of nine atoms from QM9. In this experiment, we consider two different settings. In the first one, we condition the samples to be molecules of seven Carbons, one Oxygen and one Nitrogen. In the second one, we do not enforce any assumption. As explained in §3.2.3, in this particular graph generative scattering network, the decoders for both vertices and edges are MLP's. We form them with a single hidden layer, with size 4 for vertices and 32 for edges. We take the encoder to be a graph scattering transform, with output dimension reduced to 15. In order to minimize (4), we use the Adam optimizer with a learning rate of 0.001, where we train 1,000 epochs. The computation of the scattering transform for the whole training data takes 74.04s and it takes 0.424s to train each epoch.

Using SCAT, we generate 10k molecules and record the validity, uniqueness, and novelty in Table 3. We illustrate samples generated by SCAT in Appendix §6.2. We also record in this table results reported in De Cao & Kipf (2018) for the following methods: CharacterVAE (Gómez-Bombarelli et al., 2018), GrammarVAE (Kusner et al., 2017), GraphVAE (Simonovsky & Komodakis, 2018) and MolGAN (De Cao & Kipf, 2018). Since there is no publicly available implementation of these algorithms, we cannot test their results on our machine using our setting.

The settings of Simonovsky & Komodakis (2018), De Cao & Kipf (2018) and SCAT are not the same. Simonovsky & Komodakis (2018) assumes nine atoms for each molecule and sets aside 10k samples for testing and 10k for validation. De Cao & Kipf (2018) consider molecules with no more than nine atoms and use padding if the number is less than nine. They also use the full dataset for training. We cannot compare these methods in our setting. We are also unable to test SCAT in the setting of Simonovsky & Komodakis (2018) since their training set is unknown to us. We are able to test SCAT in the setting of De Cao & Kipf (2018) and report its result in Table 3 as "SCAT with padding". In this setting, we use the complete QM9 dataset for training. The one-hot vectors are in $\mathbb{R}^5$ as an additional dimension is assigned for a non-existing atom due to padding. We form the single hidden layer, with size 5 for vertices and size 32 for edges. We take the encoder to be a graph scattering transform, with output dimension reduced to 15. In order to minimize (4), we use the Adam optimizer with a learning rate of 0.001, where we train 1,000 epochs. The computation of the scattering transform for the whole training data takes 110.57s and it takes 0.842s to train each epoch.

Our network achieves high scores for validity and novelty and moderate score for uniqueness. The moderate values of the uniqueness score may indicate on some form of mode collapse. That is, the generated molecules may not well-represent the whole variety of molecules, but a certain mode of their distribution. Nevertheless, the only two methods with higher uniqueness modes, CharacterVAE and GraphVAE, have low scores of validity and their scores of novelty are not sufficiently competitive.

Table 3: Graph generation comparison of SCAT and other algorithms using the QM9 dataset. Values are reported in percentages. The results of the other methods are taken from De Cao & Kipf (2018); Simonovsky & Komodakis (2018) since their codes are not publicly available. We remark that SCAT (conditioned) and SCAT (unconditioned) report results trained on 90k subset of data, while SCAT with padding report results trained on the complete data; all comparisons are according to 10k generated samples.

| Algorithm | Valid | Unique | Novel |
|---|---|---|---|
| CharacterVAE | 10.3 | 67.5 | 90.0 |
| GrammarVAE | 60.2 | 9.3 | 80.9 |
| GraphVAE | 55.7 | 76.0 | 61.6 |
| GraphVAE (imp) | 56.2 | 42.0 | 75.8 |
| GraphVAE (NoGM) | 81.0 | 24.1 | 61.0 |
| MolGAN | 98.1 | 10.4 | 94.2 |
| **SCAT (conditioned)** | 98.5 | 24.8 | 99.0 |
| **SCAT (unconditioned)** | 92.6 | 19.3 | 99.8 |
| **SCAT with padding** | 93.9 | 17.6 | 98.6 |

## 5 CONCLUSION

We proposed the graph generative scattering network as a generative model for graphs and graph signals. The network allows a prescribed encoder which does not require training and is approximately invariant to permutations and stable to graph deformations. Numerical experiments show competitive results for the tasks of link prediction in citation data and molecule generation. We believe this method has the potential to be used in a wider range of applications on graphs.

We used the Fashion-MNIST dataset as a sanity check for the graph signal generation, since we are unaware of a more convincing application for this specific task. In this application, we do not expect graph-based methods to compete with general methods, because graphs only retain partial spatial relationships. Indeed, the resolution of the generated images is not as good as that of the original images. Nevertheless, since the results in the similar discrete tasks of link prediction and molecule generation are competitive, we believe that SCAT also bears promise for graph signal generation, when the signals are of low resolution.

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

## 6 APPENDIX

### 6.1 SIGNAL GENERATION USING GRAPH VERSIONS OF VAE AND GAN

In §4.2 we demonstrated "boots" generated by SCAT. In this section we generate images using a graph version of VAE and GAN. The graph structure is identical to that in §4.2.

We first construct a graph VAE using the graph convolutional layers by Kipf & Welling (2017). The encoder of the VAE consists of two graph convolutional layers, while the decoder consists of two fully-connected layers. The latent mean and variation both have dimensions 256. The VAE is trained using Adam with learning rate 0.001 and 200 epochs. To compare with §4.2, the reconstruction error is 0.252, or 0.144% of the L1 norm of the original data. Note that the random generation by VAE also suffers from loss of high-frequency information.

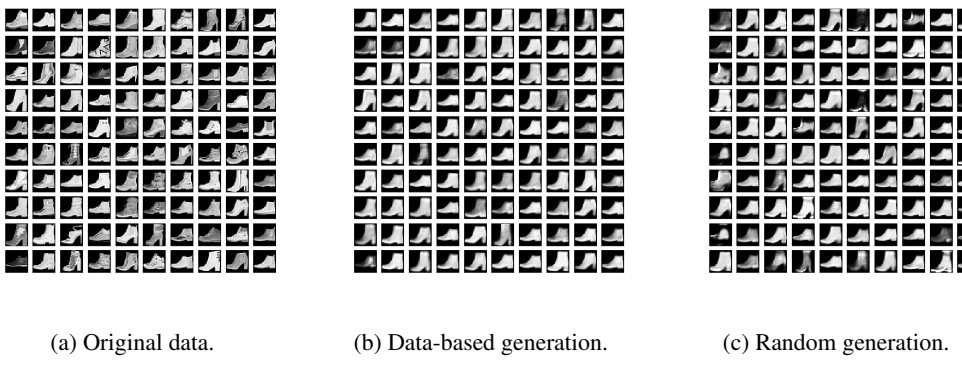

(a) Original data.          (b) Data-based generation.          (c) Random generation.

Figure 6: Original and VAE-generated images for boots of MNIST-fashion data. Figure 6a shows samples from the original data, 6b shows samples generated using features encoded by the VAE from the same samples, and Figure 6c shows samples generated from random Gaussian noise using the trained VAE.

We also construct a graph GAN. Since it is unclear how to optimally use the graph structure for GAN, we replace the discriminator with two graph convolutional layers by Kipf & Welling (2017) and use two fully-connected layers for the generator. we call this model GAN-GCN. For comparison, we also use a vanilla GAN composed purely of fully connected networks, which we call GAN-FC. The dimension of the noise is 256 for both GAN-GCN and GAN-FC. We train both models using Adam with learning rate 0.001 and 200 epochs. We observe that the discriminator in GAN-GCN converges too fast and discourages the training of the generator. GAN-FC produces more reasonable samples, but still suffers from loss of high-frequency information. Note that there is no reconstruction loss to report for GANs.

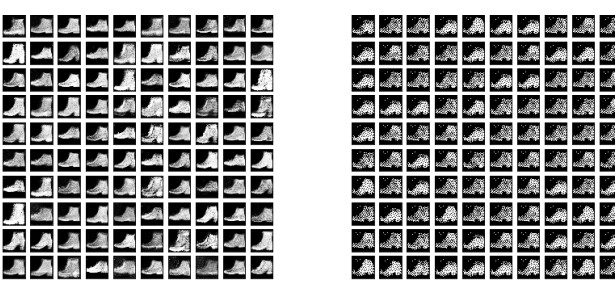

(a) Generation using GAN-FC.          (b) Generation using GAN-GCN.

Figure 7: Generated images for boots of MNIST-fashion data using GANs.

### 6.2 MOLECULE GENERATION EXAMPLES

Figure 8 demonstrates samples generated by SCAT for generating molecules trained on QM9. For each generated molecule it also reports the Quantitative Estimate of Drug-likeness (QED) score (Bickerton et al., 2012), which aims to quantify the likelihood that a molecule is a drug.

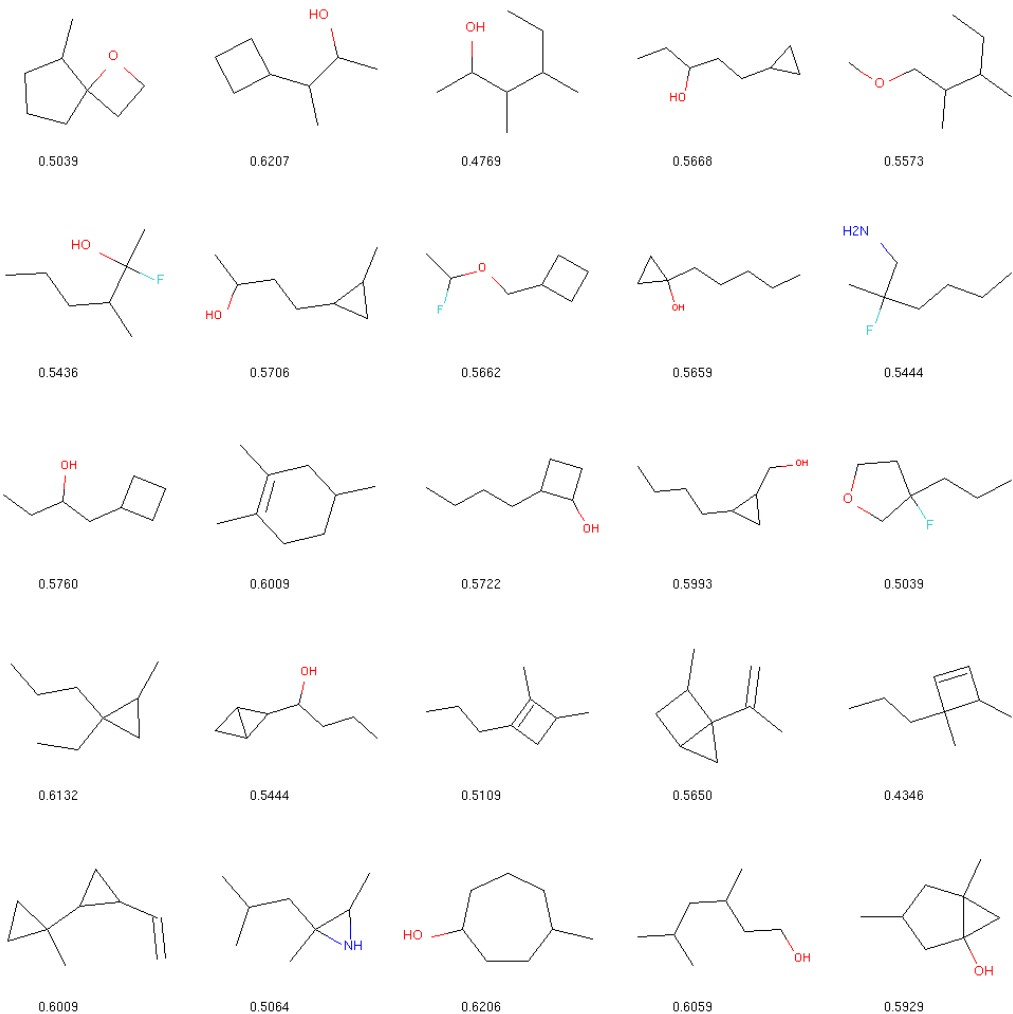

Figure 8: Generated samples by SCAT for QM9. The QED score is listed for each molecule.

