# OpenReview forum: "Graph Generation via Scattering"
_ICLR.cc/2019/Conference_

### Official Review · AnonReviewer3 · 2018-10-29
**Interesting paper, may lack novelty and clarity, seems to have problems in experimental evaluation.**

**Rating:** 4
**Confidence:** 4

**Review:**

Summary:
The paper presents a generative model for graphs which is a VAE-like architecture where the encoder is a scattering transform with fixed parameters (rather than a trainable neural net)

Pros:
+ The problem of graph generative models is very important, and a lot of the existing methods are not very scalable.
+ Using spectral methods in place of standard "neural net operations" makes a lot of sense.
+ Numerical results for the "link prediction" task seem to be significantly better than those of baselines.

Cons:
- The paper contains various imprecisions (see the non-exhaustive list below), and significant amount of statements which are hard to understand.
- I am not sure if the work can be considered particularly novel: in particular, it is not really emphasised what is the difference with [Angles & Mallat '2018].
- The motivation for the work is not entirely clear: it is true that GANS and VAEs have their issues, but in my view it is not really explained / argued why the proposed method would solve them.
- I find the argument about the efficiency not very convincing, especially after looking at the members (bottom of p. 7): the scattering transform alone takes several orders of magnitude longer than the baseline. Authors also mention that their method does not require training of the encoder, but I do not see any comparisons with respect to number of parameters.
- The experimental evaluation for "signal generation" and "graph generation" is not very convincing. For the former there is no real comparison to existing models. And for the latter, the experimental setup seems a bit strange: it appears that the models were trained on different subsets of the dataset, making the comparison not very meaningful. Also, I would expect to see the same methods to be compared to a cross all the tasks (unless it is impossible for some reason).

Various typos / imprecisions / unclear statements:
p.1, "are complex as well as difficult to train and fine-tune.": not at all clear what this means.
p.1, "Their development is based on fruitful methods of deep learning in the Euclidean domain, such as convolutional and recurrent neural networks.": Recurrent and convolution neural network are not necessarily restricted to Euclidean domains.
p.1, "Using a prescribed graph representation, it is possible to avoid training
the two components at the same time, but the quality of the prescribed representation is important
in order to generate promising results.": not clear what this sentence means.
p.2, "Unlike GAN or VAE, the model in this paper does not require training two components either iteratively or at the same time.": I do not see why that would necessarily be a bad thing, especially in the case of VAE where traditional training in practice corresponds to training a single neural net.
p.3, "GAN-type graph networks use a discriminator in order to compete with the generator and make its training more powerful.": I am not sure this statement is strictly correct.
p.9: "We remark that the number of molecules in the training sets are not identical to that in ...": does this mean that the models are effectively trained on different data? In that case, the comparison is not very meaningful.

---

> ### Author Response · Authors · 2018-11-27
> **Response to Reviewer 3**
>
>
> We believe that, as indicated by reviewer 2, the paper is clear and well-written. Later on, we address all the small issues you raised. Nonetheless, we think it is unfair to claim that there are various (or possibly many) imprecisions and significant amount of statements which are hard to understand. We agree that there were few unclear sentences.
>
> We discussed issues with novelty above. As for the particular comparison with Angles & Mallat (2018). We indeed follow the basic framework of this paper, but we use it with the recent graph scattering transform instead of the original one, we suggest it for different graph tasks with different types of decoders and most importantly, we are able to obtain competitive results for the discrete graph applications, unlike the results of Angles and Mallat for image data, which are not competitive with state-of-the-art generative neural networks.
>
> In comparison to GANs and VAEs the proposed method has the following advantages: 1. There is no need to train the encoder. 2. There is an established mathematical understanding of the robustness of the encoder to signal and graph manipulation (see Zou & Lerman (2018)). 3. The numerical results of this method are more competitive.
>
> The scattering transform is not part of the training and thus requires to be executed only once. In the numerical section we train 1000 epochs and thus the total time (25.75s for Cora and 17.76s for Citeseer) is smaller than that for VGAE (80s for Cora and 108.2s for Citeseer). Even if we just train 200 epochs (as in the VGAE paper), the total time is still smaller than that for VGAE. In particular, application on our machine of VGAE to the PubMed dataset exhausted all computing resources. Therefore, there is a benefit for not training the encoder.
>
> We indeed claim that our method does not require training of the encoder. As for parameters chosen for the scattering transform, they are very generic and cannot be considered as requiring training. We use the Shannon wavelets, which are the simplest wavelets and have no special parameters, we choose J = 3 and 3 layers, similarly to Zou & Lerman (2018) and we do not expect any improvement with higher J or additional layers. Also, choices of reduced dimensions are mentioned in the text. Dimension reductions were performed in order to save time and reduce redundancies. We did not notice sensitivity with different dimensions chosen. There are no hidden parameters that need to be carefully tuned.
>
> Our numerical part emphasizes the graph generation and not the signal generation. The signal generation is presented as a simple sanity check and there are no quantitative estimates for it. Also, one needs to recall that image generation models produce better images than graph-based generation models. There was no space to include images generated by other methods and such a comparison may not be meaningful. We comment though that we did similar experiments with GAN and VAE and we now report them in the appendix. Since we are not aware of any previous work addressing this task, we constructed the networks by replacing specific parts in a standard GAN / VAE with graph networks. For GAN, the discriminator is a graph neural net following Kipf & Welling (2017) and the generator is fully-connected; for VAE, the encoder is a graph neural net following Kipf & Welling (2017). It seems that GAN is worse and VAE is comparable, but it is hard to quantify the differences.
>
> Both MolGAN and GraphVAE do not have their codes available online. We thus had to use their reported results. Therefore, we cannot have the same training set as GraphVAE (their training set was not specified). Furthermore, GraphVAE used a validation set, where other works do not need it. Also, GraphVAE fixes the number of atoms to be nine for each molecule and thus does not use training data with less than 9 atoms. On the other hand, MolGAN trains over the whole QM9 dataset. It also requires padding to deal with fewer atoms than 9. Our particular choices make sense to us and there is nothing we could do about exact comparison with their codes as they did not provide them. We made it very clear in the original manuscript. We do not see any way to exactly compare with the original setting of GraphVAE, but we can compare with the exact setting of MolGAN. Even though this setting is not natural to us, we added such a comparison in the revised version. In this setting, validity by scattering is slightly lower than MolGAN but still good, on the other hand, the novelty and uniqueness by scattering are higher.

---

> > ### Author Response · Authors · 2018-11-27
> > **Response to Reviewer 3 (cont'd)**
> >
> > Clarification of statements:
> >
> > * Explanation of ”are complex as well as difficult to train and fine-tune.”
> > We meant by “complex” that it is rather difficult to understand properties of GANs and VAEs. On the other hand, as we mentioned above, for our procedure there is some understanding of its robustness to signal and graph manipulation. As for “difficult to train and fine-tune”, it is well known that it is difficult to train a GAN since it suffers from local minima and diminishing gradients. Furthermore, training VAE in the Euclidean domain to produce unblurred samples is also known to be difficult. Anyway, we slightly rewrote the text.
> >
> > * Explanation of reference to the Euclidean domain.
> > We first discussed images and texts and thus talked about the Euclidean domain and later mentioned generalization to the graph domain. We slightly rewrote the text to make it even clearer. We remark though that a recurrent network assumes a sequence or time series, which has a 1D Euclidean structure. Similarly, a convolutional neural network assumes a domain where convolution makes sense. Of course there are generalized notions, as we mention, such as “graph convolution”, which is not a mathematical convolution.
> >
> > * Explanation of the importance of the quality of the prescribed representation.
> > We choose to fix a prescribed encoder and train a decoder only. In this way the loss function is purely associated with the decoder, which is easier to understand and is more tractable. However, since we do not train the encoder, it is very important to make sure that the encoding is meaningful. Properties that make our encoder meaningful are described in Zou & Lerman (2018). We rewrote this sentence.
> >
> > * On whether training two parts is a bad thing.
> > For VAE, the model can be seen as a single neural net, but its loss function corresponds to two components:
> > the encoder (represented in the KL part) and the the decoder (represented in the cross-entropy part). The main issue is how to weigh the two parts of the energy function. One way to look at the loss function is to regard it as an approximation to the Evidence Lower Bound (ELBO) and use equal weights, but it is based on a crude approximation. For GAN, training two parts is more difficult since the gradient has to be taken iteratively for the generator and the discriminator. Of course, training one component does not necessarily outperforms VAE or GAN (see e.g., the results in Angles & Mallat (2018)), but it is an interesting paradigm, whose properties might be better understood.
> >
> > * On correctness of ”GAN-type graph networks use a discriminator in order to compete with the generator and make its training more powerful.”
> > Mathematically, GANs minimize KL divergences between the data distribution and the generation distribution. However, intuitively, the discriminator is used for making the task of generation hard, thus the generator has to be sufficiently good in order to fool the discriminator.
> >
> > * On comparison of the QM9 dataset: we mentioned this issue above.

---

> > > ### Comment · AnonReviewer3 · 2018-12-05
> > > **thanks for the comments**
> > >
> > > Thanks a lot for an elaborate reply.
> > > I still believe that the paper clarity could be improved, and, generally, I am not very convinced by the arguments regarding the novelty and the benefits wrt to existing generative models.
> > > BTW, the code of MolGAN seems to be publicly available (I do not know if it was when the paper was submitted though) https://github.com/nicola-decao/MolGAN

---

> > > > ### Author Response · Authors · 2018-12-05
> > > > **Response to feedback**
> > > >
> > > > * The new uploaded version of the paper addressed your comments on the very few unclear sentences. If you have additional comments let us know.
> > > > * Thanks for pointing to the new github page with the code of MolGAN. It was not available before submission (it was only initiated, with partial information, 3 days before submission).

---

### Official Review · AnonReviewer1 · 2018-11-02
**Interesting topic, but the paper does not contain enough novel content.**

**Rating:** 4
**Confidence:** 3

**Review:**

## Summary ##

The authors apply the wavelet scattering transform to construct an autoencoder for graphs. They apply this architecture to reconstructing citation graphs, images, and generating molecules.

## Assessment ##

It was difficult to discern what parts of this paper were new work. The graph scattering transform seems to have appeared first in Hammond et al. or Zhou and Lerman. The proposed decoder in 3.2.1 is attributed to Kipf and Welling. The molecule generation portion was interesting, but I don't think there was enough novel content in this paper to justify acceptance to ICLR. I could be convinced otherwise if the authors' contribution is clarified in rebuttal.

## Questions and Concerns ##

* I found the definition of $S[p]f$ (page 5) a little confusing. In particular, what constitutes a 'path' $p$ in this setting?
* Can you motivate the whitening operation $A$ that is applied to the encoding? It seems like this is eliminating a lot of the information encoded in $\bar{X}$.
* I'm confused by the choice of loss function at the top of page 6. Since $D(z) = \sigma(...)$, it seems like $D(i, j)$ is meant to represent the probability of a link between $i$ and $j$. In that case, the loss is a sum of negative probabilities, which is unusual. Was this meant to be a sum of log probabilities? Also, this loss doesn't seem to account for including edges where there are none. Can you explain why this is the case?
* In section 4.2, the encoded dimension is 256 IIUC. Considering that the data was restricted to the "boots" class, the reduction from 784-->256 dimensions does not seem significant. The authors concede that some high-frequency information is lost, so I wonder how their approach compares to e.g. a low-pass filter or simple compression algorithm.
* Section 4.3 states that the molecules considered are constructed from atoms C, H, O, N and F. Later, there are multiple references to only 4 atom types, one-hot vectors in $R^4$ etc. Clarify please!

---

> ### Author Response · Authors · 2018-11-27
> **Response to Reviewer 1**
>
>
> We believe that we give clear credits to previous works and it is thus surprising that “it is difficult to discern what parts of this paper were new work.” As stated in the paper the graph scattering transform is due to Zou and Lerman and it relies on the graph wavelets of Hammond et al. We claimed that “the adjustment of the structure of the decoder to the three types of tasks does not require a lot of effort”. The specific decoder we used for link prediction is a very natural choice (just like using a natural fully connected network for convolutional neural networks). This decoder is also not a main issue in Kipf & Welling’s work (but the idea of combining VAE and graph convolution for graph tasks makes it interesting). Despite the fact that we follow previous works, we believe they are several interesting points to people who care about graph generation. We detailed these points above when addressing the novelty concern of reviewer 2.
>
> Specific responses:
>
> * In the definition of S[p]f (page 5) a “path” p is a sequence of scales (indeed, it was denoted in the text by p = (j_1, · · · , j_m) on Page 5). Each scale corresponds to a wavelet transform at a certain layer (we further clarify it in the new manuscript). Figure 1 in Zou & Lerman (2018) clarifies the transform.
>
> * Explanation of the whitening operation A and possible elimination of information encoded in \bar{X}: In generation tasks, it is common to generate a sample from Gaussian noise and send it to the decoder. Our whitening procedure maps the latent variable \bar{X} to Gaussian noise. It also reduces the dimension of the signal. We find such dimension
> reduction useful and even necessary. Indeed, the dimension of the output of the scattering transform is very high, since this output corresponds to different paths and thus has a lot of redundancy.
>
> * On the choice of loss function at the top of page 6: Thanks for noticing a typo in our paper. It should indeed be a sum of log-likelihoods and this is how we implemented the code. We guess that by “this loss doesn’t seem to account for including edges where there are none” you mean that the sum term does not include the case where W(i, j) = 0. Note that this is exactly the correct form of a cross-entropy loss. Indeed, here W(i, j) \neq 0 corresponds to probability (of connecting two vertices) 1 and W(i, j) = 0 corresponds to probability 0. A cross-entropy loss contains a term with respect to W(i, j) = 0, which is equal to 0, since the true probability is 0.
>
> * On significance of dimension reduction: The dimension is reduced to 256, not from 784, but from the dimension of the output of the scattering transform, which is, 784 × 13 = 10, 192 (note that there are 13 paths). The comment “I wonder how their approach compares to e.g., a low-pass filter or simple compression algorithm” is unclear to us.
> Please let us know what kind of comparison you would like us to pursue and what should be interesting about it.
>
> * Clarification of 4 types of atoms: It is sufficient to encode C, N, O, F and the types of bonds that connect them. It is not necessary to encode H because it will be uniquely determined by the other atoms and bonds. For instance, if we have two C’s connected by a double bond, then it has to be CH2 = CH2 (here the = sign denotes a double bond and not an equality).

---

### Official Review · AnonReviewer2 · 2018-11-03
**Simple combination of existing works**

**Rating:** 4
**Confidence:** 4

**Review:**

The paper used the graph scattering network as the encoder, and MLP as the decoder to generate links/graph signals/graphs.

Pros:
1.	Clearly written. Easy to follow.
2.	No need to train the encoder
3.	Good results on link prediction tasks

Cons:
1.	Lack of novelty. It is a simple combination of existing encoders and decoders. For example, compared to VGAE, the only difference in the link prediction task is using a different encoder. Even if the performance is very good, it can only demonstrate the effectiveness of others’ encoder work and this paper’s correct selection of a good encoder.
2.	Lack of insights. As a combination of existing works, if the paper can deeply explain the why this encoder is effective for the generation, it is also beneficial. But we also do not see this part. In particular, in the graph generation task, the more important component may be the decoder to regulate the validness of the generated graphs (e.g. “Constrained Generation of Semantically Valid Graphs via Regularizing Variational Autoencoders. In NIPS 2018” which used the similar decoder but adding strong regularizations in VAE).
3.	 Results on QM9 not good enough and lack of references. Some recent works (e.g. “Junction Tree Variational Autoencoder for Molecular Graph Generation, ICML 2018”) could already achieve 100% valid.

---

> ### Author Response · Authors · 2018-11-27
> **Response to Reviewer 2**
>
>
> On lack of novelty:
>
> Even though the components of the graph scattering generative network are derived from previous works, it raises several interesting points: 1. It describes a universal way of using graph scattering for different graph generation tasks. 2. Unlike GAN and VAE based methods, the encoding phase does not require training. 3. Unlike the generative scattering transform of Angles and Mallat (2018), which does not perform as well as state-of-the-art methods for imaging tasks, the proposed one is very competitive for discrete graph data and it should be noted. To demonstrate the problem with your criticism that you expressed as “simple combination of existing encoders and decoders”, one may apply it to the very interesting work of Angles and Mallat in ICLR 2018 and misjudge its contribution.
>
> On lack of insights:
>
> The reason why this generation method is useful can be explained by its robustness to signal and graph manipulation (see Proposition 5.1 and Theorems 5.2 and 5.3 in Zou and Lerman (2018)). Nevertheless, in our opinion, practical performance, which is emphasized here, is more important to verify (we mention above practical deficiencies of the Euclidean generative network of Angles and Mallat (2018)).
>
> On QM9 performance and lack of reference:
>
> Even though we find JT-VAE, the method in the ICML paper you mentioned, interesting and we now refer to it, it is not directly relevant to our experiments. Also, your claim “could already achieve 100% valid” is not precise. JT-VAE achieves 93.5% validity without validity check in the decoding phase and 100% only after a full validity check (please check the conference version and not the arXiv one). More importantly, the result is on a different dataset (ZINC), not QM9. For QM9, the tree decomposition of JT-VAE is not expected to improve results since molecules in QM9 are composed of at most nine atoms. Moreover, JT-VAE reinforces molecule validity (see e.g., Step 3 in Algorithm 1 of the ICML paper). This validity reinforcement can be applied to other graph-based methods and to be fair to other methods, comparison with JT-VAE should include applying it to them too.

---

### Meta-Review · Area_Chair1 · 2018-12-14
**Some merit.**

**Confidence:** 4
**Recommendation:** Reject

**Metareview:**

AR1 is concerned about the novelty and what are exact novel elements of the proposed approach. AR2 is worried about the novelty (combination of existing blocks) and lack of insights. AR3 is also concerned about the novelty, complexity and poor  evaluations/lack of thorough comparisons with other baselines. After rebuttal, the reviewers remained unconvinced e.g. AR3 still would like to see why the proposed method would be any better than GAN-based approaches.

With regret, at this point, the AC cannot accept this paper but AC encourages the authors to take all reviews into consideration and improve their manuscript accordingly. Matters such as complexity (perhaps scattering networks aren't the most friendly here), clear insights and strong comparisons to generative approaches are needed.